# Insecticide Resistance of *Cimex lectularius* L. Populations and the Performance of Selected Neonicotinoid-Pyrethroid Mixture Sprays and an Inorganic Dust

**DOI:** 10.3390/insects14020133

**Published:** 2023-01-27

**Authors:** Jin-Jia Yu, Sabita Ranabhat, Changlu Wang

**Affiliations:** Department of Entomology, Rutgers—The State University of New Jersey, 96 Lipman Dr., New Brunswick, NJ 08901, USA

**Keywords:** topical assay, insecticide resistance, neonicotinoid-pyrethroid mixture, inorganic dust

## Abstract

**Simple Summary:**

The common bed bug (*Cimex lectularius* L.) experienced a worldwide resurgence during the last two decades. Chemical control is one of the popular strategies to control bed bug infestations. Bed bugs developed resistance to insecticides, but the prevalence and levels of insecticide resistance among local bed bug populations and whether the commonly used insecticide products are effective on these populations, is unclear. This study tested 13 field populations of bed bugs collected in the United States and determined their insecticidal resistance levels. We found that seven populations developed very high level resistance to deltamethrin. Only one population exhibited high level resistance to neonicotinoids. The performance of three neonicotinoid-pyrethroid mixture sprays and an inorganic insecticide dust were evaluated using three field populations that exhibited high levels of resistance to deltamethrin. All three tested populations required much higher concentrations (55–2017 times higher) than the laboratory strain to induce 90% mortality. Exposure to silica gel dust caused >95% mortality after 72 h. These findings indicate most resistant bed bugs may not be effectively controlled by pyrethroids, neonicotinoid, or pyrethroid-neonicotinoid insecticide sprays. Insecticide dusts containing silica gel are an effective material for control of bed bug infestations.

**Abstract:**

Insecticide resistance is one of the factors contributing to the resurgence of the common bed bug, *Cimex lectularius* L. This study aimed to profile the resistance levels of field-collected *C. lectularius* populations to two neonicotinoids and one pyrethroid insecticide and the performance of selected insecticide sprays and an inorganic dust. The susceptibility of 13 field-collected *C. lectularius* populations from the United States to acetamiprid, imidacloprid, and deltamethrin was assessed by topical application using a discriminating dose (10 × LD_90_ of the respective chemical against a laboratory strain). The RR_50_ based on KT_50_ values for acetamiprid and imidacloprid ranged from 1.0–4.7 except for the Linden 2019 population which had RR_50_ of ≥ 76.9. Seven populations had RR_50_ values of > 160 for deltamethrin. The performance of three insecticide mixture sprays and an inorganic dust were evaluated against three *C. lectularius* field populations. The performance ratio of Transport GHP (acetamiprid + bifenthrin), Temprid SC (imidacloprid + β-cyfluthrin), and Tandem (thiamethoxam + λ-cyhalothrin) based on LC_90_ were 900–2017, 55–129, and 100–196, respectively. Five minute exposure to CimeXa (92.1% amorphous silica) caused > 95% mortality to all populations at 72 h post-treatment.

## 1. Introduction

The common bed bug (*Cimex lectularius* L.) and the tropical bed bug (*C. hemipterus* F.) are hematophagous ectoparasites that have been recognized as pests throughout human history [1]. Common bed bugs are mainly found in temperate climate regions whereas tropical bed bugs are mainly restricted to tropical areas. Due to the application of modern synthetic insecticides, bed bugs had become uncommon for decades. However, bed bug resurgence occurred around the world in the last two decades [2,3,4,5,6,7]. Various factors have contributed to their resurgence including lack of public awareness, exchange of second-hand items including furniture, increased local and international travel, changes in pest management practices, and resistance to insecticides. Among these, insecticide resistance has been cited as the main reason contributing to the resurgence of bed bugs [8,9,10,11,12,13].

In the United States, pyrethroids are widely used by consumers as well as professionals for general pest control, with the inclusion of bed bugs. Among 245 interviewed residents who used insecticides to treat bed bug infestations, 72% used pyrethroids [14]. Among the top ten insecticide products used by pest management professionals in the U.S., eight are pyrethroids or contain pyrethroids [15]. Pyrethroids are sodium channel modulators, causing uninterrupted nerve firing, which results in insects exhibiting shaking and rapid death [16]. The predominant use of pyrethroid-based products has selected for resistance in many populations of bed bugs in different parts of the world [11,12,17,18,19,20]. Cross-resistance with DDT, which was previously widely used to control bed bug infestations, may have also contributed to high levels of resistance in modern day bed bugs [17]. Enhancement of the detoxification enzymes [21,22,23,24] and insensitivity on target sites (*kdr* mutation) [9,19,25,26] are the most studied mechanisms for pyrethroid resistance. Further, cuticular modification was also confirmed in a pyrethroid-resistant bed bug strain [27] which may reduce the penetration of insecticide into insect’s body.

Over past decade, neonicotinoids have been combined with pyrethroids into formulated insecticides for bed bug management, especially for controlling pyrethroid resistant populations [28,29,30]. Neonicotinoids mimic the agonist action of acetylcholine at nicotinic acetylcholine receptors that result in overstimulation of the nervous system causing involuntary muscle contraction, cessation of feeding, paralysis, and ultimately leads to death [16,31]. In general, neonicotinoid-pyrethroid mixtures perform better than pyrethroid-only insecticides [32] due to the dual modes of action. Previous laboratory and field studies showed outstanding performance of neonicotinoid-pyrethroid mixture sprays for bed bug control. Transport (acetamiprid + bifenthrin) exhibited ≥ 89.7% residual efficacy on all tested surfaces (fabric, unpainted wood, painted wood, and vinyl), whereas pyrethroids were much less effective on all tested substrates when tested against a moderately resistant field strain bed bugs [32]. Two non-chemical methods (installing encasements and applying steam) plus insecticide mixture spray application, namely Tandem (thiamethoxam + λ-cyhalothrin), Temprid SC (imidacloprid + β-cyfluthrin), and Transport (acetamiprid + bifenthrin), caused ≥ 87% *C. lectularius* trap count reduction after eight weeks in a field study [29].

There are only several reports of high resistance to neonicotinoids. Romero and Anderson [33] reported acetamiprid and imidacloprid resistance on *C. lectularius* collected from Michigan and Cincinnati in the U.S. Campbell and Miller [34] pointed out that *C. lectularius* collected from Richmond and Cincinnati were resistant to neonicotinoid-pyrethroid sprays. Dang et al. [35] also found that field-collected *C. hemipterus* have developed various levels resistance to Temprid SC and Tandem (thiamethoxam + λ-cyhalothrin). Additionally, Gordon et al. [18] reported an increased resistance of bed bug toward Temprid SC and Transport GHP insecticides after one-generation selection. These studies indicate that the selection pressure from neonicotinoid-pyrethroid mixtures will and have already caused resistance development among field bed bug populations.

Besides insecticide sprays, dust formulations are also commonly used by professionals in the U.S. for the management of bed bugs [15]. One of the advantages of insecticide dusts is that dusts are more readily picked up by bed bugs compared to dry spray residue and a potential horizontal transfer effect [36]. Most of the commercial dust formulations contain pyrethroid or pyrethrin as active ingredients for bed bug management. However, pyrethroids have a repellent effect on insects and the existing pyrethroid resistance will affect the efficacy of dusts in bed bug management [37]. Consequently, inorganic dusts including diatomaceous earth (DE) dust and silica gel have been recommended to control bed bug populations [38,39]. DE dust and silica gel kill bed bugs by removing the wax layer from the cuticle of an insect thus causing desiccation [40,41,42]. Singh et al. [43] reported silica gel dust being the most effective among eight commonly used dusts including DE dust. Further, Ranabhat and Wang [44] reported that silica gel dust performed better than neonicotinoid-inorganic mixture (0.25% dinotefuran + 99.75% DE) and pyrethroid dust (1% cyfluthrin) against *C. lectularius*. According to the previous studies, silica gel dust may be a viable alternative material for the control of resistant bed bugs.

In spite of the fact that insecticide resistance is widespread and has been reported as the main factor responsible for bed bug resurgence, chemical control remains the most common approach in managing bed bugs. Therefore, monitoring insecticide resistance and performance of formulated insecticides is critical for designing the most effective bed bug management programs. In 2019, we did preliminary studies and the results showed that continuous exposure to Suspend SC (4.75% deltamethrin) dry residue caused 2–92% mortality (median value = 18%) to seven field *C. lectularius* populations collected in New Jersey. Exposure to Home Defense (0.05% bifenthrin, 0.0125% cypermethrin) dry residue caused 0.4–74% mortality (median value = 69%) to four field *C. lectularius* populations (three from New Jersey and one from Indiana). Among these 11 field populations, seven of them exhibited < 90% mortality after exposure to Transport GHP dry residue continuously. To provide a more accurate resistance profile of the field bed bug populations, we conducted the current studies to: (1) assess the resistance levels of *C. lectularius* populations to acetamiprid, imidacloprid, and deltamethrin using topical bioassays; and (2) to evaluate the performance of three commonly used neonicotinoid-pyrethroid mixture sprays and a silica gel dust product.

## 2. Material and Methods

### 2.1. Bed Bug Populations

A total of 13 field-collected populations and one laboratory strain of *C. lectularius* were used in this study (Appendix A). Fort Dix is a susceptible laboratory strain that has never been exposed to insecticides since 1973 [20,45]. Aberdeen, Dehart, and Irvington 624-5G populations were collected from a single home or apartment. The rest of the bed bug populations were collected from multiple apartments within the same apartment building. All bed bugs were maintained in plastic containers (5.0 cm diameter and 4.7 cm height, Consolidated plastics, Stow, OH, USA) and several pieces of red folded card paper were provided as harborages. Bed bugs were fed defibrinated rabbit blood (Hemostat Laboratories, Dixon, CA, USA) every 2–4 weeks using a Hemotek membrane-feeding system (Discovery Workshops, Accrington, UK). All bed bug populations were kept in an environmental chamber at 25 ± 1 °C, 45 ± 10% RH, and a photoperiod of 12:12 (L:D) h. Bed bugs were fed to repletion 3 to 7 d prior to all bioassays.

### 2.2. Resistance to Two Neonicotinoids and a Pyrethroid Insecticide Using Topical Assays

Three technical grade insecticides belonging to neonicotinoid and pyrethroid groups were tested: acetamiprid (99.5%), imidacloprid (98.9%), and deltamethrin (93.2%). All materials were purchased from Chem Service Inc., West Chester, PA, USA. In topical assays, a feather weight forceps (available at www.amazon.com, accessed on 27 December 2022) were used to place 10 male adults of unknown age in a clean Petri dish (5.5 cm diameter and 1.5 cm height, Fisher Scientific, Pittston, PA, USA) lined with filter paper (Grade P8, Fisher Scientific, Pittston, PA, USA). Only male bed bugs were used in the topical assays because female bed bugs suffer injuries from traumatic insemination during mating. The wound on the abdomen of females may affect the results. Bed bugs were then anesthetized by placing the Petri dish on ice for 1 min prior to topical assay. An insecticide-acetone solution (1 µL) was applied onto the dorsal surface of the abdomen using a micro-applicator (Burkard Manufacturing Co. Ltd., Rickmansworth, UK) equipped with a 25-µL glass syringe (Burkard Manufacturing Co. Ltd., Rickmansworth, UK). Treated bed bugs were held with a forceps until the applied insecticide solution had dried then gently placed in a clean Petri dish. Control insects received 1 µL of acetone only. The concentrations of insecticides used were acetamiprid: 0.1–50 ppm; imidacloprid: 1–10 ppm; and deltamethrin: 0.04–1 ppm. At least three concentrations causing 1–99% mortality were included for each chemical. Each concentration was replicated three times. After treatment, insects were maintained in Petri dishes in an environmental chamber at 25 ± 1 °C, 45 ± 10% RH and 12:12 L:D photoperiod. Mortality was recorded after 72 h. An insect was considered dead if it could not move when touched with a pair of forceps.

After calculating the values of lethal dose (LD) for each insecticide, a discriminating dose (10 times of LD_90_; 10 × LD_90_) [46] of three insecticides was tested against all bed bug populations using topical assay. Similar to the previous topical assays, ten males were transferred to a clean Petri dish lined with filter paper. Each insect was treated with 1 µL insecticide-acetone solution. Insects in the control were treated with 1 µL acetone. Each population was replicated three times. The number of knockdown and dead insects was recorded every 5 min for the first hour and subsequently every 10 min until 2 h or until 90% of tested insects were knocked down. A bed bug was considered knocked down if it could not right itself when gently flipped over with a pair of forceps. Final mortality was recorded after 72 h. All tests were conducted in 2022.

### 2.3. Performance of Formulated Insecticides

#### 2.3.1. Neonicotinoid-Pyrethroid Mixture Sprays

Three insecticide sprays were evaluated against three deltamethrin resistant populations: Bayonne 2015, Irvington, and Irvington 624-5G. The insecticides included Transport GHP (FMC Corporation, Philadelphia, PA, USA), Temprid SC (Bayer Crop Science LP, Research Triangle Park, NC, USA), and Tandem (Syngenta Crop Protection, Greensboro, NC, USA) (Appendix A). All three formulations contain a neonicotinoid and a pyrethroid insecticide. A stock solution with the concentration 10 times higher than the label rate was made for each insecticide. Preliminary tests were conducted to obtain the range of concentration for each spray. After determining the range, 5 to 6 different concentrations of each spray were made by serial dilutions with tap water. A Potter Spray Tower (Burkard Scientific Ltd., Herts, UK) was used to apply the sprays on a Petri dish (5.5 cm diameter) lined with filter paper with a delivery rate of 4.0–4.2 mg/cm^2^ (approximately 1.0 gallon/1000 ft^2^). The control groups were treated with tap water only. Treated Petri dishes were allowed to air dry for 24 h before the tests. Ten male adults of unknown age and 10 nymphs (4th to 5th instar) from each population were introduced onto the treated paper using feather weight forceps. The Petri dishes were maintained in an environmental chamber and the mortality was checked every 24 h until 72 h. Dead bed bugs were removed during each examination. Each concentration was replicated three times. This experiment was conducted in 2019.

#### 2.3.2. Inorganic Dust

Due to the poor performance of mixture sprays on resistant bed bug populations, an inorganic dust (CimeXa, 92.1% amorphous silica, Rockwell Labs Ltd., North Kansas, MO, USA) was selected to test against Bayonne 2015, Irvington, and Irvington 624-5G populations. The dust was applied evenly to an unpainted birch plywood panel (12.54 cm × 12.54 cm, Revell Inc., Elk Grove Village, IL, USA) using a paint brush. CimeXa dust was applied at label rate of 0.61 mg/cm^2^. Twenty male adult bed bugs were placed onto the dust-treated panels using feather weight forceps and confined with a plastic ring (9 cm diameter) for 5 min. Control plywood panels did not receive any treatment. After the 5 min exposure, insects were transferred to a clean Petri dish lined with filter paper and mortality was observed at 1 h and subsequently at an interval of 24 h until 144 h. Each insecticide treatment was replicated four times. This experiment was conducted in 2021.

### 2.4. Data Analysis

Control knockdown and mortality were corrected using Abbott’s formula [47]. The control mortality was ≤ 10% for all populations in all tests. The lethal dose (LD) and knockdown time (KT) to kill 50 and 90% of the tested bed bugs in topical assays were generated using a natural log probit model in SPSS v. 11.0 (SPSS Inc., Chicago, IL, USA). The lethal concentration to kill 50% of the tested bed bugs (LC_50_) was determined by probit analysis using SAS software version 9.3 (SAS Institute, Cary, NC, USA). The resistance ratio (RR) of field bed bug populations was calculated by dividing the KT_50_ of corresponding population by the KT_50_ of Fort Dix strain. The classification of insecticide resistance followed Lee and Lee [48] study: ≤ 1 time = no resistance; > 1 to 5 time(s) = low resistance; > 5 to 10 times = moderate resistance; > 10 to 50 times = high resistance; > 50 times = very high resistance. Performance ratio values were calculated for insecticide sprays by dividing the LC_50_ of corresponding population by the LC_50_ of Fort Dix strain. One-way analysis of variance (ANOVA) with Tukey’s HSD test was used for analyzing the efficacy of dust insecticide. The data were checked for normal distribution and arsine of the square root transformed if they were not normally distributed. ANOVA was performed using SPSS v. 11.0.

## 3. Results

### 3.1. Resistance to Two Neonicotinoids and a Pyrethroid Insecticide

The lethal dose (LD_50_, LD_90_) values of three insecticides against the Fort Dix strain are shown in Table 1. Fort Dix strain showed the highest value of LD_50_ (2.64 ng/µL) and LD_90_ (18.64 ng/µL) to acetamiprid, followed by imidacloprid with LD_50_ = 2.41 ng/µL and LD_90_ = 6.73 ng/µL. Deltamethrin showed the lowest LD_50_ (0.90 ng/µL) and an intermediate value of LD_90_ (11.49 ng/µL). The discriminating doses were calculated as 10× LD_90_ values, which are 186 ng/µL for acetamiprid, 67 ng/µL for imidacloprid, and 115 ng/µL for deltamethrin. In all topical assays, control insects had no mortality after 72 h.

Topical assays using the discriminating dose of acetamiprid showed the Linden 2019 population had the highest resistance with RR_50_ of higher than 288 and RR_90_ higher than 93.9 (Table 2). The mean mortality of Linden 2019 was 13 ± 3% at 72 h post-treatment. The Aberdeen population was the 2nd most resistant population with RR_50_ of 4.1 and RR_90_ higher than 93.9. The mean mortality was 57 ± 3% after 72 h. Other populations had low detected resistance to acetamiprid with RR_50_ values ranging from 1 to 3.6 and RR_90_ ranging from 1.4 to 5.4. The mean mortality reached 100% at 72 h, except 93 ± 3% mortality for the New Brunswick population (Table 2).

For imidacloprid, only Linden 2019 population exhibited very high resistance (RR_50_ = 76.9, RR_90_ > 81.5) and a moderate mean mortality (60 ± 6%) after 72 h (Table 3). New Brunswick was the next most resistant population with RR_50_ of 4.7 and RR_90_ of 7.4. However, the mean mortality was 100% after 72 h. Other populations had low resistance with values of RR_50_ ranging from 1.2 to 3.1 and RR_90_ ranging from 1.3 to 3.3 (Table 3). 

The majority of the field-collected *C. lectularius* populations exhibited very high levels of resistance to deltamethrin. Aberdeen, Bayonne 2015, Canfield, Irvington, Irvington 624-5G, Linden 2019, and New Brunswick populations had RR_50_ greater than 160 and RR_90_ greater than 91.9 (Table 4). The mean mortality of these seven populations were less than 30% after 72 h. Cotton and Indy had low resistance to deltamethrin based on RR_50_ of 1.7 and 1.8, but had very high resistance based on RR_90_ of > 91.9 and moderate mortality occurred (73% and 67%) after 72 h. The most susceptible field populations were Bayonne, Dehart, Hackensack, and Masiello with both RR_50_ and RR_90_ ranging from 1.0 to 3.4 and the mean mortality were all higher than 90% after 72 h. 

### 3.2. Performance of Formulated Insecticides

#### 3.2.1. Neonicotinoid-Pyrethroid Mixture Sprays

In Transport GHP treatments, all field populations exhibited significant lower susceptibility to Transport GHP compared to the laboratory population (Fort Dix). The lethal concentration (LC_90_) ranged from 0.54–1.21%. The performance ratios based on LC_90_ (PR_90_) against Bayonne 2015, Irvington, and Irvington 624-5G were 1883, 900, and 2017, respectively (Table 5). In Temprid SC treatments, the LC_90_ ranged from 0.30–0.71% and the performance ratios (PR_90_) against Bayonne 2015, Irvington, and Irvington 624-5G were 129, 75, and 55, respectively. In Tandem treatments, the LC_90_ values were 0.52–1.02% and the PR_90_ against Bayonne 2015, Irvington, and Irvington 624-5G were 194, 100, and 196, respectively. In all experiments, the mean mortality in the control was 0–7%.

Among the three tested field populations, their susceptibility levels to three tested mixtures were Irvington < Irvington 624-5G < Bayonne 2015. When comparing the performance of the three insecticide mixtures, Temprid SC performed better (with lower PR_90_) than Transport GHP and Tandem. Tandem also performed better against Bayonne 2015 and Irvington 624-5G (with lower PR_90_) than Transport GHP.

#### 3.2.2. Inorganic Dust

CimeXa was highly effective against the laboratory strain (Fort Dix), causing 92% and 100% mortality at 24 h and 48 h after treatment (Figure 1). Among the field populations in the CimeXa treatment, Irvington 624-5G exhibited slower mortality than the laboratory strain at 24 h and 48 h. However, all field populations reached > 95% mortality at 72 h post-treatment. There was no mortality in control treatment.

## 4. Discussion

In the current study, a total of 13 *C. lectularius* populations were collected from 10 cities in the U.S. The collection sites were regularly treated with insecticides by pest control contractors or licensed housing staff supplemented by residents’ self-treatment using pyrethroids or essential oil-based products. Bayonne and Indy populations were collected prior to insecticide mixture products becoming widely available and used for bed bug management. The frequent insecticide treatment may explain the various levels of resistance to neonicotinoids and deltamethrin in the current study. For the two neonicotinoids tested, most bed bug populations had low resistance. Only the Linden 2019 population evolved with very high level resistance to both acetamiprid and imidacloprid. Additionally, the Aberdeen population had very high level resistance to acetamiprid based on RR_90_ (> 93.9) and mortality of 57%. In another study, all three tested bed bug populations were found moderately to highly resistant to imidacloprid and acetamiprid [33]. The general esterase, glutathione S-transferase, and cytochrome P450 (CYP450) monooxygenase enzyme activity were found enhanced in neonicotinoid-resistant populations. 

This study used 10× LD_90_ value of a lab strain as discriminating dose based on previous studies. After this study was completed, we tested 8 field bed bug populations using 5× LD_90_ of Fort Dix strain of acetamiprid to confirm the validity of 10× LD_90_ as discriminating dose. The results showed that RR_50_ values were 1.7–3.9 (median 2.5) based on 5× dose compared to 1.0–4.1 (median 2.0) based on 10× dose. Six out of eight populations had slightly higher RR_50_ values than those based on 10× dose. However, seven out of eight populations had higher RR_90_ values than those based on 10× dose. The RR_90_ values were 3.0–41.0 (median 7.9) based on 5× dose compared to 1.4–> 93.9 (median 2.6) based on 10× dose. Furthermore, one additional population (Canfield) exhibited a < 100% mortality when tested with 5× dose. Hence, both 5× and 10× could be used for detecting resistance but using 5× dose may reveal higher RR_90_ values.

Previous studies have documented bed bug resistance to different pyrethroids including deltamethrin [49,50,51,52], λ-cyhalothrin [49,52,53], β-cyfluthrin [33], and permethrin [54]. In the current study, 69% (9 out of 13) of field-collected bed bug populations evolved with high RR_90_ (> 91.9). Deltamethrin resistance in bed bug field populations from the U.S. was mentioned in previous studies [26,55]. Cotton and Indy populations exhibited RR_50_ for deltamethrin less than 2.0, but the mortality was only 73% and 67%, respectively. Similarly, Leong et al. [56] found that bed bugs treated with Tandem insecticide showed a RR_50_ value of 1.4 but the mortality was only 53%. The key mechanisms causing pyrethroid resistance among bed bugs are overexpression of detoxification enzymes and the target site insensitivity (*kdr*-type mutation) [57,58]. The heterozygous *kdr* mutation of bed bug population may affect knockdown response and lead to low mortality [25]. 

The bed bug populations used in this study were collected from 2008 to 2021 (Appendix A). It was suggested a possible energy trade-off between insecticide resistance development and life history parameters will cause the resistance of populations to return to susceptible rapidly [30]. However, nine field-collected populations still exhibited high deltamethrin resistance in the current study. One population (Irvington, Appendix A) had a RR_50_ > 160 after being maintained in the laboratory for approximately 10 yr. Likewise, *C. hemipterus* exhibited high resistance (RR_50_ > 600) to neonicotinoid-pyrethroid mixture insecticides after a 14 yr maintenance without exposure to insecticides [56]. The bed bug resistance to insecticides may therefore not degrade as rapidly as aforementioned. 

In previous studies, neonicotinoids remain effective against bed bugs even though they had high pyrethroid resistance [12,13,59]. However, the current results suggest neonicotinoid-based insecticides may not be able to effectively control many bed bug populations. The label rate for Tandem SC, Temprid SC, and Transport GHP is 0.13, 0.075, and 0.11%, respectively. Based on the LC_90_ values as shown in Table 5, none of these products would be able to cause > 90% mortality to the three tested field *C. lectularius* populations after 3 d, suggesting potential control difficulties using these products. Although further observations beyond 3 d may reveal additional mortality, the majority of the mortality occurred within 3 d after treatment in our previous experiments. Interestingly, topical assays showed low level resistance to neonicotinoids in the three populations. Thus, the very high PR of the three insecticide sprays may primarily be due to high deltamethrin resistance and even the presence of low level resistance to neonicotinoids may render the neonicotinoid–pyrethroid mixture spray treatment ineffective. Similarly, field-collected *C. lectularius* and *C. hemipterus* populations were found resistant to Temprid SC, Transport GHP, and Tandem [34,60]. Upregulated CYP450 enzymes was reported as the main mechanism for bed bug deltamethrin resistance [22,61] and can also detoxify neonicotinoid insecticides [33,62,63]. The mechanisms causing the poor performance of neonicotinoid-pyrethroid mixture in the three *C. lectularius* populations requires further investigation in future studies.

CimeXa is a silicon dioxide-based (SiO_2_) desiccant dust which can absorb the lipids from insects’ cuticle and cause them to dehydrate and die [64,65]. In laboratory studies, CimeXa caused high mortality (> 95%) to bed bug within 24 h [43,66]. In a field treatment, CimeXa application plus non-chemical methods led to an approximately 60% reduction in bed bug populations within one month [67]. Lilly et al. [68] also showed that bed bug strain possessed *kdr*-type resistance exhibited 100% mortality at 72 h after CimeXa treatment. Similarly in the current study, CimeXa caused high mortality (> 95%) to bed bugs after 72 h, even in populations with high deltamethrin-resistance. However, significantly delayed mortality was observed in Irvington 624-5G population at 24 h and 48 h compared with Fort Dix strain. The modified cuticle may explain the delayed mortality. Increased cuticular thickness [27,69] was found in resistant bed bugs, therefore it may contribute to the delayed response of insect to insecticides [70,71]. Whether these mechanisms are responsible for the delayed response to CimeXa remains to be determined. 

In conclusions, the majority of the field-collected *C. lectularius* populations exhibited high levels of resistance to deltamethrin. Very high resistance to neonicotinoids was only found in one of the 13 field populations studied. However, the performance of the insecticide mixtures varied, and all three tested *C. lectularius* populations showed much lower susceptibility to neonicotinoid-pyrethroid sprays than the laboratory strain, suggesting control difficulties may occur using these insecticides in the field. Inorganic insecticide dust containing silica gel provided high mortality (> 95% at 72 h) against pyrethroid-resistant bed bug populations. According to the current study, applying inorganic insecticide dust is an effective option in managing resistant bed bug populations.

## Figures and Tables

**Figure 1 insects-14-00133-f001:**
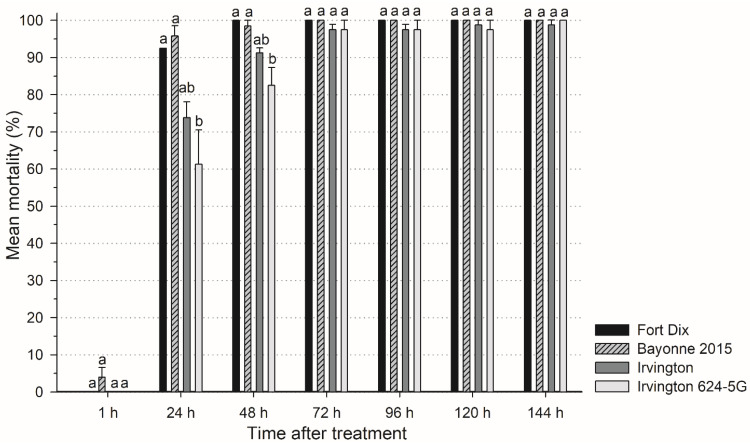
Mean mortality (%) of laboratory *C. lectularius* and three field-collected *C. lectularius* populations exposed to CimeXa insecticide for 5 min on unpainted wood panel. For each observation period, the mean mortality with same letter within same time means no significant difference among populations (ANOVA, *p* > 0.05). Control groups had no mortality.

**Table 1 insects-14-00133-t001:** Lethal dose (LD_50_, LD_90_) value of three insecticides against a laboratory strain *C. lectularius* using topical assays.

Insecticide	* N	Model Parameters		Lethal Dose (ng/μL)		Model Fit		
		** Intercept ± SE	Slope ± SE	LD_50_ (95% CI)	LD_90_ (95% CI)	χ^2^	df	*p*
Acetamiprid	150	−0.636 ± 0.162	1.509 ± 0.223	2.64 (1.78–3.80)	18.64 (11.10–43.56)	4.207	3	0.240
Imidacloprid	150	−1.096 ± 0.220	2.873 ± 0.390	2.41 (1.92–2.94)	6.73 (5.21–9.87)	2.151	3	0.542
Deltamethrin	150	0.053 ± 0.188	1.156 ± 0.264	0.90 (0.55–2.78)	11.49 (3.42–250.84)	0.327	3	0.955

* Total number of treated insects with 30 insects in each concentration. ** The intercept and slope parameters are for models in which the independent variable is natural logarithm of dose.

**Table 2 insects-14-00133-t002:** Susceptibility (based on knockdown time, KT_50_ and KT_90_) of 13 field-collected *C. lectularius* populations to topically applied acetamiprid at a discriminating dose of 186 ng per adult.

Population	n	KT_50_ (min, 95% CI)	KT_90_ (min, 95% CI)	χ^2^ (df)	Slope ± SE	% Mean Mortality (±SE) at 72 h	* RR_50_	RR_90_
Fort Dix	30	15 (13–18)	46 (39–56)	9.2 (11)	2.692 ± 0.266	100 ± 0	-	-
Aberdeen	30	61 (51–76)	>4320	6.9 (12)	1.694 ± 0.210	57 ± 3	4.1	>93.9
Bayonne	30	37 (31–42)	141 (110–200)	4.8 (11)	2.191 ± 0.220	100 ± 0	2.5	3.1
Bayonne 2015	30	16 (12–19)	80 (57–140)	4.4 (7)	1.814 ± 0.257	100 ± 0	1.0	1.7
Canfield	30	21 (17–24)	75 (61–97)	7.4 (12)	2.290 ± 0.222	100 ± 0	1.4	1.6
Cotton	30	32 (28–39)	126 (94–200)	4.4 (10)	2.159 ± 0.258	100 ± 0	2.1	2.7
Dehart	30	11 (7–14)	65 (46–118)	3.6 (7)	1.636 ± 0.262	100 ± 0	1.0	1.4
Hackensack	30	28 (23–33)	71 (54–119)	15.5 (9)	3.125 ± 0.360	100 ± 0	1.9	1.5
Indy	30	39 (35–44)	115 (92–159)	10.9 (12)	2.716 ± 0.288	100 ± 0	2.6	2.5
Irvington	30	16 (8–24)	82 (47–375)	48.4 (11)	1.804 ± 0.207	100 ± 0	1.1	1.8
Irvington 624-5G	30	26 (22–30)	126 (96–188)	14.1 (13)	1.868 ± 0.204	100 ± 0	1.7	2.7
Linden 2019	30	>4320	>4320	-	-	13 ± 3	>288	>93.9
Masiello	30	43 (36–50)	239 (174–382)	6.2 (16)	1.711 ± 0.181	100 ± 0	2.9	5.2
New Brunswick	30	54 (46–65)	249 (175–430)	5.1 (10)	1.934 ± 0.232	93 ± 3	3.6	5.4

* RR (resistance ratio based on KT): KT of field population/KT of Fort Dix strain.

**Table 3 insects-14-00133-t003:** Susceptibility (based on knockdown time, KT_50_ and KT_90_) of 13 field-collected *C. lectularius* populations to topically applied imidacloprid at a discriminating dose of 67 ng per adult.

Population	n	KT_50_ (min, 95% CI)	KT_90_ (min, 95% CI)	χ^2^ (df)	Slope ± SE	% Mean Mortality (±SE) at 72 h	* RR_50_	RR_90_
Fort Dix	30	16 (11–20)	53 (42–77)	12.1 (7)	2.445 ± 0.317	100 ± 0	-	-
Aberdeen	30	50 (41–62)	136 (98–255)	32.0 (13)	2.928 ± 0.291	100 ± 0	3.1	2.6
Bayonne	30	38 (31–48)	122 (87–218)	14.8 (9)	2.532 ± 0.264	100 ± 0	2.4	2.3
Bayonne 2015	30	20 (16–24)	80 (60–127)	5.7 (6)	2.108 ± 0.272	100 ± 0	1.3	1.5
Canfield	30	19 (16–22)	69 (52–112)	10.3 (7)	2.255 ± 0.309	100 ± 0	1.2	1.3
Cotton	30	26 (21–33)	93 (65–179)	15.5 (9)	2.344 ± 0.263	100 ± 0	1.6	1.8
Dehart	30	34 (30–40)	95 (73–144)	7.8 (7)	2.876 ± 0.365	100 ± 0	2.1	1.8
Hackensack	30	23 (21–26)	53 (43–71)	6.9 (6)	3.627 ± 0.458	100 ± 0	1.4	1.0
Indy	30	38 (31–50)	126 (85–259)	1.5 (5)	2.465 ± 0.381	100 ± 0	2.4	2.4
Irvington	30	19 (16–22)	68 (54–94)	11.2 (9)	2.305 ± 0.263	100 ± 0	1.2	1.3
Irvington 624-5G	30	39 (33–46)	174 (123–302)	15.7 (11)	1.960 ± 0.240	100 ± 0	2.4	3.3
Linden 2019	30	1231 (760–2361)	>4320	10.2 (15)	0.856 ± 0.087	60 ± 6	76.9	>81.5
Masiello	20	35 (30–40)	102 (80–148)	6.1 (11)	2.728 ± 0.332	100 ± 0	2.2	1.9
New Brunswick	20	75 (58–108)	394 (223–1194)	4.1 (8)	1.775 ± 0.307	100 ± 0	4.7	7.4

* RR (resistance ratio based on KT): KT of field population/KT of Fort Dix strain.

**Table 4 insects-14-00133-t004:** Susceptibility (based on knockdown time, KT_50_ KT_90_) of 13 field-collected *C. lectularius* populations to topically applied deltamethrin at a discriminating dose of 115 ng per adult.

Population	n	KT_50_ (min, 95% CI)	KT_90_ (min, 95% CI)	χ^2^ (df)	Slope ± SE	% Mean Mortality at 72 h	* RR_50_	RR_90_
Fort Dix	30	27 (24–29)	47 (43–53)	4.8 (10)	5.206 ± 0.467	100	-	-
Aberdeen	30	>4320	>4320	-	-	17 ± 7	>160	>91.9
Bayonne	30	44 (39–48)	86 (75–105)	2.0 (9)	4.294 ± 0.438	97 ± 3	1.6	1.8
Bayonne 2015	30	>4320	>4320	-	-	13 ± 9	>160	>91.9
Canfield	30	>4320	>4320	-	-	20 ± 6	>160	>91.9
Cotton	30	44 (377–52)	>4320	4.7 (17)	1.508 ± 0.166	73 ± 9	1.6	>91.9
Dehart	30	24 (21–27)	55 (47–69)	2.7 (7)	3.593 ± 0.394	93 ± 3	1.0	1.2
Hackensack	30	27 (21–34)	56 (41–120)	19.3 (7)	4.015 ± 0.493	100	1.0	1.2
Indy	30	48 (40–64)	>4320	3.6 (10)	1.699 ± 0.253	67 ± 7	1.8	>91.9
Irvington	30	>4320	>4320	-	-	27 ± 7	>160	>91.9
Irvington 624-5G	30	>4320	>4320	-	-	0	>160	>91.9
Linden 2019	30	>4320	>4320	-	-	3 ± 3	>160	>91.9
Masiello	30	54 (46–59)	162 (137–202)	9.9 (19)	2.687 ± 0.217	90 ± 10	2.0	3.4
New Brunswick	30	>4320	>4320	-	-	7 ± 7	>160	>91.9

* RR (resistance ratio based on KT): KT of field population / KT of Fort Dix strain.

**Table 5 insects-14-00133-t005:** Performance of three formulated insecticide mixture products against one laboratory strain (Fort Dix) and three field-collected *C. lectularius* populations. Bed bug morality was recorded at 72 h.

Spray	Population	Slope (±SE)	Lethal Concentration (%)		Performance Ratio		Model Fit	
			* LC_50_ (95% CI)	LC_90_ (95% CI)	** PR_50_	PR_90_	χ^2^	df
Transport GHP	Fort Dix	2.9 ± 0.4	0.0002 (0.0002–0.0003) a	0.0006 (0.0005–0.0009) a	-	-	2.6	4
	Bayonne 2015	2.1 ± 0.3	0.51 (0.31–0.67) b	1.13 (0.93–1.56) b	2550	1883	9.3	4
	Irvington	1.1 ± 0.3	0.04 (0.01–0.08) c	0.54 (0.25–4.29) bc	200	900	9.6	4
	Irvington 624-5G	1.7 ± 0.2	0.45 (0.35–0.54) b	1.21 (1.08–1.40) b	2250	2017	19.1	4
Temprid SC	Fort Dix	1.9 ± 0.4	0.0012 (0.0005–0.0023) a	0.0055 (0.0028–0.0365) a	-	-	8.2	4
	Bayonne 2015	2.0 ± 0.3	0.16 (0.12–0.22) b	0.71 (0.47–1.39) b	133	129	13.7	4
	Irvington	1.1 ± 0.2	0.03 (0.01–0.07) c	0.41 (0.13–15.30) bc	25	75	9.3	4
	Irvington 624-5G	2.6 ± 0.3	0.10 (0.08–0.12) d	0.30 (0.23–0.41) c	83	55	4.2	4
Tandem	Fort Dix	1.2 ± 0.1	0.0004 (0.0003–0.0006) a	0.0052 (0.0032–0.0096) a	-	-	5.3	4
	Bayonne 2015	2.5 ± 0.4	0.58 (0.45–0.75) b	1.01 (0.88–1.57) b	1450	194	8.5	4
	Irvington	1.6 ± 0.2	0.08 (0.52–0.11) c	0.52 (0.33–1.01) b	200	100	4.9	4
	Irvington 624-5G	2.0 ± 0.0	0.24 (0.19–0.30) d	1.02 (0.75–1.59) b	600	196	3.2	4

* LC_50_ or LC_90_ values of different populations followed different letters are significantly different based on non-overlapping values. ** PR (performance ratio based on LC): LC of resistant population/LC of Fort Dix population.

## Data Availability

The data presented in this study are available in this article.

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
