# Peer review of "Insecticide Resistance of Cimex lectularius L. Populations and the Performance of Selected Neonicotinoid-Pyrethroid Mixture Sprays and an Inorganic Dust"

_insects, 2023, doi:10.3390/insects14020133_

Round 1

Reviewer 1 Report

1. Could you explain in your materials and method to suggest why only males were included in the experiment?

2. Why do you choose these 3 particulars insecticide formulations (Transport, Tandem, and Temprid) as your testing subjects? There are also other insecticide formulations that were labelled for bed bug treatment, for instance a few formulations from Sumitomo Chemical.

3. Could you explain why did you use 10x of LD99 as your discrminating dosage? Noticed that you compared the validity of using 10x by conducting another set of experiment with 5x. Why not using the LD90 as the discriminating dose? As there might be a risk of masking the actual resistance level or underestimate the resistance status of a population when using higher discriminating dosage.

4. Noticed that you include both RR50 and RR90 in your results. I do noticed that there is a few populations that exhibited large differences between RR50 and RR90. What do you think causes the differences? Could you include a few sentences in your manuscript to explain which of it do you think is more suitable to represent the resistance status of a population?

Reviewer 2 Report

I have the following comments/changes

1. Line 24 Insecticide dusts containing silica gel is an effective material for control of bed bug infesta- 24 

change is to are

2. In the introduction line 80 change three to two and take out namely

Three non-chemical methods (installing encasements 80 and applying steam) plus insecticide mixture spray application namely, namely

3. In line 91 take out status

 Additionally, Gordon et al. [18] reported an increased resistance status of bed bug toward 

4. In line 104 change and to thus and cause to causing

silica gel kill bed bugs by removing the wax 103 layer from the cuticle of an insect and cause desiccation

5. In line 117 take out the U.S. and add a period after Jersey

New 116 Jersey, the U.S. Exposure to Home Defense (

In materials and methods

6. In line 167 take out it.

Line: flipped it over with 

7. In line 205 take change were to was

Line: bed bugs (LC50) were determined by

Results

8. in line 221-222 add a comma after acetamidprid, lowercase follow, and change following to followed.  Take out value of 

Line: acetamiprid. Following by imidacloprid with LD50 = 2.41 ng/μl and LD90 = 221 6.73 ng/μl. Deltamethrin showed the lowest value of LD50 

9. In line 266 change ranging to ranged

Line concentration (LC90) ranging from 0.54-1.21%.

10.  In line 272 take out between

Line: In all experiments, the mean mortality in the control was between 0-7%.

11. In line 320 take out been.

Line: Previous studies have been documented

12. Line 329-330 take out the and a

Line: bed bug population may affect the knockdown response and lead to a low mortality

13.  In line 333 add to after populations and to after return

Line: populations return suscepti-333 ble rapidly [30]. However, nine field-collected populations still exhibited

14. In line 340 change to againist.

line: In previous studies, neonicotinoids remain effective to bed bugs even though they

15. In line 370 take out the.  

Line: layed the response of insect to the insecticide 

Reviewer 3 Report

Dear authors

Congratulations by their study; where determined the resitant populations of Cimex lectularius to diferent insecticides..
